# TOWARDS GOOD PRACTICE IN BOOSTING THE TARGETED ADVERSARIAL ATTACK

## ABSTRACT

By accessing only the surrogate model, attackers can craft adversarial perturbations to fool black-box victim models into misclassifying a given image into the target class. However, the misalignment between surrogate models and victim models raises concerns about defining what constitutes a successful targeted attack in a black-box setting. In our work, we empirically identify that the vision-language foundation model CLIP is a natural good indicator to evaluate a good transferable targeted attacks. We find that a successful transferable targeted attack not only confuse the model on the vision modality towards the target class, but also fool the model on the text modality between the original class and target class. Motivated by this finding, we propose a simple yet effective regularization term to boost the existing transferable targeted attacks. We also revisit the feature-based attacks, and propose to boost the performance by enhancing the fine-grained features. Extensive experiments on the ImageNet-1k dataset demonstrate the effectiveness of our proposed methods. We hope our finding can motivate future research on the understanding of targeted attacks and develop more powerful techniques.

## 1 INTRODUCTION

While deep neural networks have achieved remarkable progress across various applications, their vulnerability to adversarial examples has raised significant concerns regarding the reliability of their practical deployment. Adversaries can craft targeted attacks to generate imperceptible perturbations and add them to benign samples, manipulating the decisions of these models. Moreover, the existence of adversarial transferability enables the application of adversarial examples generated on white-box surrogate models to efficiently attack black-box models as well. Exploring methods to enhance the transferability of targeted adversarial attacks can provide valuable insights into the nature of adversarial examples and motivate the design of robust learning techniques for trustworthy AI applications.

Targeted attacks are more challenging than untargeted ones because they require the crafted perturbation to not only confuse the neural networks but also to misclassify the object as a specific target class. Several studies have explored transferable targeted adversarial attacks, focusing on designing advanced objectives, input transformations, and feature- and model-based attack methods. These methods optimize the perturbation based on feedback from the neural networks to achieve the target class. For instance, Zhao et al. (2021) and Weng et al. (2023) propose maximizing the logit of the target class, while Inkawhich et al. (2019b), Gao et al. (2021), and Inkawhich et al. (2019a) suggest amplifying the image features associated with the target class, *etc.* Despite these strategies and tricks designed to enhance performance, a natural question arises: *What is the key factor contributing to successful targeted attacks in a black-box setting?*

To answer this question, we first need criteria to justify the success of black-box targeted attacks in general. As shown in fig. 1, while previous work usually evaluates targeted attack performance on a limited number of victim models, it fails to establish a reliable criterion when there is no knowledge about the dataset used to train the victim models, *e.g.*, the potential misalignment in the definition of the target class between the surrogate

model and the target model in an open-world setting. To address this problem, we propose leveraging CLIP as a fair indicator to evaluate the effectiveness of different targeted attack methods. There are two main reasons for using CLIP: First, CLIP is a foundation model trained on a large-scale dataset, making it more robust than conventional models. Second, CLIP is trained on hundreds of billions of image-text pairs, creating a more interpretable latent space that combines vision and language modalities. A successful targeted attack in a black-box setting should not only be robust enough to transfer to various models but also closely align with the target class in both vision and text modalities.

In this work, we empirically identify that the CLIP model without fine-tuning is a naturally good indicator to evaluate a successful targeted attack under the practical black-box setting. Motivated by this finding, we propose to distill the direction towards the targeted class during attack and design a simple yet effective regularization term to further boost the performance of existing powerful transferable attack methods. Besides, during the evaluation process, we find the feature-based attacks always achieve the spurious performance compared with others, which motivates us to deep dive into its effectiveness. We also conduct empiri-

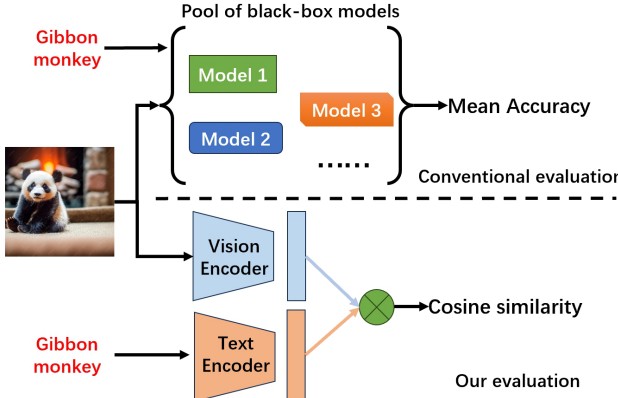

Figure 1: CLIP is a natural good indicator to model the target class information and evaluate the target adversarial attack performance under the black-box setting.

cal study on the key factor contibuting to its success, and propose to leverage the fine-graiend features to further improve the performance. Extensive experiments on Imagenet-1K dataset verify the effectiveness of our method.

Our contributions are summarized as follows,

- We propose a new metric based on the CLIP to identify the effectiveness of the transferable adversarial attack.
- We design a simple yet effective regularization term to enhance the existing transferable adversarial attacks.
- We empirically identify the key factors contributing to the effectiveness of the feature-based attack, and propose to leverage more fine-grained features to boost the performance.
- We conduct experiments on ImageNet-1K dataset to validate the effectiveness of our proposed method.

## 2 RELATED WORK

### 2.1 TARGETED TRANSFERABLE ADVERSARIAL ATTACK

Szegedy et al. (2014) first identifies the existence of adversarial examples, which are crafted by adding the human-imperceptible perturbation on benign samples to fool models' decisions. Targeted transferable adversarial attacks are the most threatening in real-world applications, which target manipulating the black-box models' decisions at attackers' will. Many methods have been proposed for targeted transferable adversarial attacks, which can be generally categorized into four kinds, namely the input transformation-, advanced objective-, feature- and model-based methods.

**Input transformation-based methods** advocate exploiting the input diversity for better generalization ability in the optimization process, thus improving the adversarial transferabil-

ity. While the methods studying the untargeted adversarial attacks can be directly used in the context of targeted attacks, such as DIM, TIM, and Admix, there are some novel input transformation-based methods mainly studying the targeted attacks. For example, Byun et al. (2022) diverse the inputs based on 3D objects to enhance the targeted adversarial attacks. Liu & Lyu (2024) propose a local mix-up strategy to randomly mix regions of transformed adversarial images with each other, thus boosting the input diversity towards better targeted adversarial attack performance.

**Advanced objective-based methods** design specialized loss functions to boost targeted adversarial attacks. Li et al. (2020) introduce the Poincare distance as the similarity metric to make the gradient more self-adaptive and suitable in the context of targeted attacks, thus alleviating the curing of inherent noise in decreasing the transferability. Zhao et al. (2021) propose the logit loss to directly enlarge the logit output of the targeted class to boost the targeted attack performance. Weng et al. (2023) proposes to increase the logit margin to deal with the saturation problem for better targeted adversarial transferability.

**Feature-based methods** focus on optimizing the latent space of adversarial images to improve the targeted adversarial transferability. Wei et al. (2023) enhances the targeted adversarial transferability by maximizing the similarity between the latent features of original images and cropped images. Inkawhich et al. (2019b) and Gao et al. (2021) optimize the feature of adversarial examples towards that of the image from target class. Inkawhich et al. (2019a) attack the image by maximizing its posterior probability of the features for the target class. Byun et al. (2023) fuses features of other benign samples with those of adversarial examples to boost the targeted adversarial transferability,

**Model-based methods** train better surrogate models or directly use the generative model to generate adversarial perturbation for targeted adversarial attacks. Springer et al. (2021) find that a surrogate model that is more robust to adversarial perturbation can be leveraged to craft adversarial examples with highly targeted adversarial transferability. Yang et al. (2022a) propose a hierarchical generative network to generate targeted adversarial perturbation to fool neural networks. Wang et al. (2023) design a generative adversarial training framework for targeted attacks, which consists of a generator used for crafting targeted adversarial examples, and feature-label dual discriminators to detect the adversarial examples from the images of the target class.

## 2.2 ADVERSARIAL DEFENSE

Several strategies have been proposed to mitigate the risk of adversarial attacks on neural networks. These include adversarial training (Madry et al., 2018; Tramèr et al., 2018; Wang et al., 2021), input preprocessing (Xie et al., 2018; Naseer et al., 2020), feature denoising (Liao et al., 2018; Xie et al., 2019; Yang et al., 2022b), and certified defenses (Raghunathan et al., 2018; Gowal et al., 2019; Cohen et al., 2019), among others. Liao et al. (2018) developed a denoising autoencoder, referred to as the High-level Representation Guided Denoiser (HGD), which aims to cleanse adversarial perturbations. Xie et al. (2018) introduced a technique that involves random resizing of the image and adding padding to reduce adversarial effects, called Randomized Resizing and Padding (R&P). Xu et al. (2018) proposed the Bit Depth Reduction (Bit-Red) method, which reduces the bit depth per pixel to mitigate perturbations. Liu et al. (2019) defended against adversarial attacks using a JPEG-based compression method on adversarial images. Cohen et al. (2019) employed randomized smoothing (RS) to train a certifiably robust classifier. Naseer et al. (2020) proposed a Neural Representation Purifier (NRP) designed to eliminate perturbations. We use defense methods to evaluate the performance of targeted adversarial attacks.

## 3 METHODOLOGY

**Notations**. Given the image $x$ with the label $y$, the attacker can generate the human-imperceptible adversarial perturbation $\delta$, which leads the image classifier $f$ to misclassify $x$ into the targeted class $\hat{y}$. The optimization of $\delta$ can be formulated as follows,

$$\arg\min_{\delta} \mathcal{L}(f(x+\delta), \hat{y})), \quad s.t. \ \|\delta\|_{\infty} < \epsilon, \tag{1}$$

where $\mathcal{L}$ is the classification loss, *e.g.*, the cross-entropy function, and $\epsilon$ is the maximum perturbation budget under the $L_{\infty}$ norm constraint. Many studies have identified the existence of adversarial transferability, where the adversarial examples generated by the surrogate model $f$ can be used to fool other black-box models.

**Settings**. In this study, we explore the key factors contributing to a successful transferable targeted attack. In our default setting, we randomly choose $1,000$ images from the ImageNet-1K dataset as our evaluation set, which are classified by our tested model. The targeted attacked classes are also randomly generated. We use eight surrogate/victim models, comprising 1) Convolutional Neural Network (CNNs): ResNet-18, ResNet-101, ResNXt-50, and DenseNet-121; and 2) transformers: ViT, PiT, Visformer, and Swin. We generate adversarial examples using different surrogate models and evaluate their performance on all tested models, *i.e.*, under both the white- and black-box setting. We set the maximum perturbation magnitude $\epsilon = \frac{16}{255}$ under the $L_{\infty}$ constraint. Unless otherwise specified, we set the number of iterations as 300 (Zhao et al., 2021), the step size as $\frac{2}{255}$.

### 3.1 Overview of Three Popular Methods and Beyond

We start the discussion from studying three popular targeted attack methods, which achieves the state-of-the-art performance in recent years, and generally followed by others.

**Logit** (Zhao et al., 2021) directly maximizes the logit output of the target class using a large number of iterations, achieving superior target transferability. While it is simple to implement and effective compared to optimizing the loss shown in Eq. (1) (*Pros.*), it overlooks the competition class and the original class, leading to performance degradation in targeted transferability (*Cons.*).

**Logit-margin** (Weng et al., 2023) builds upon the Logit attack by scaling the logits with a temperature factor and an adaptive margin, which is the difference between the top-1 and top-2 logits. Additionally, it reveals that minimizing the cosine similarity between the input feature of the final classification layer and the classifier weights of the target category can improve transferability. This method benefits from considering the competition class (*Pros.*), showing significant improvement over the Logit method. However, it is still limited by under-explored targeted features and exhibits poor transferability across different black-box models (*Cons.*).

**CFM** (Clean Feature Mixup) (Byun et al., 2023) is a targeted adversarial attack method based on feature fusion. It pre-computes the clean features of benign samples and randomly mixes them with the features of adversarial examples during the attack process. The diverse features introduced encourage the attack to explore more alternative optimization directions on the landscape, thus achieving an effective and efficient targeted transferable attack (*Pros.*). However, focusing solely on the feature space of the targeted class limits its potential for performance improvement (*Cons.*).

By utilizing the target class information of the surrogate model from different levels, *i.e.*, the logit at the most abstracting bottom level to the features at the top level, we observe consistent improvements in attack performance under the black-box setting (see table 2) and derive the following assumption: *the key to successful transferable targeted attacks is to fully utilize the generalized information of the target class to amplify robust target class features while alleviating competition class features, where the competition class is typically the original class.*.

### 3.2 Representing the target class by CLIP

*How can the information belonging to the target class be represented more generally?* The aforementioned three methods utilize the surrogate model itself. However, as identified by previous studies, different models share similar regions of interest but differ in their decision boundaries, which affects adversarial transferability. This difference makes targeted attacks

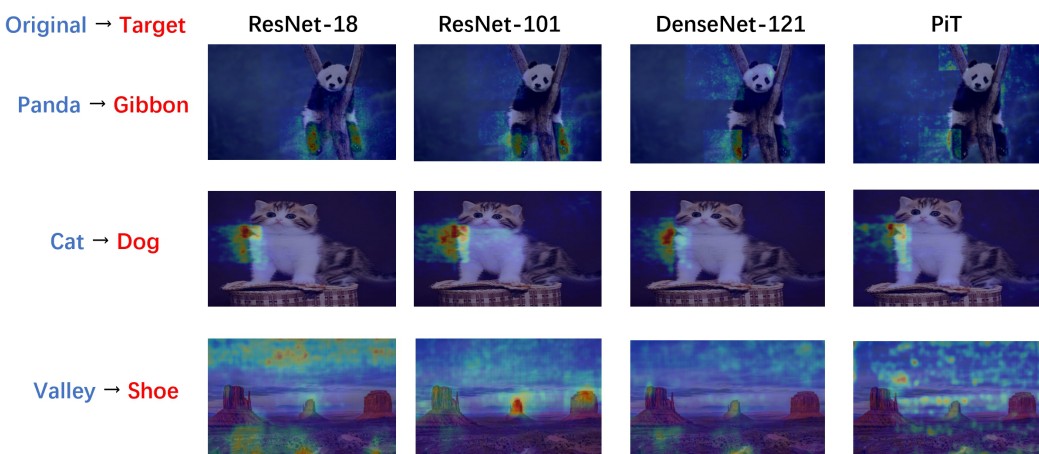

Figure 2: We use GradCAM with SESS to visualize the regions of interest for different models when they misclassify images into various target classes. The red areas indicate the most important features contributing to the model's decision, while the blue areas indicate the least important features.

in a black-box setting more challenging. While untargeted attack methods focus on features irrelevant to the original class, targeted attack methods need to concentrate on the features specific to the target class, which are more precise and vary significantly among different models. These factors make it difficult to model the target class information using only the local single surrogate model.

To better understand the difficulty of targeted adversarial transferability, we use saliency maps to visualize the impact of different features on model decisions for specific classes. To more precisely reflect the contribution of local features, we enhance the performance of the saliency map with SESS. The results are presented in fig. 2. As seen, when there is a large difference between the original class and the target class, the regions of interest for the model show more conflicts across different models. For example, the region of interest for changing a "cat" to a "dog" remains relatively consistent, while it differs significantly when changing a "valley" to a "shoe." Different models have significantly different preferences in misclassified decisions, which can hinder the success of targeted transferable attacks.

To better represent the target class information, we propose leveraging CLIP, which is trained on hundreds of billions of text-image pairs, resulting in more robust representations in the latent space. Compared with conventional deep learning models, CLIP uses contrastive learning, allowing the target class information to be represented jointly by text and vision modalities. While collecting images to access the general image features of the target class is challenging, obtaining text representations is easier. These text representations are entangled with potential image features in the latent space. Thus, it is intuitive to use the cosine similarity between the features of the image and the text embedding as the distance to the targeted class.

To validate the reasonability of using CLIP to model the target class information, we first generate 1,000 adversarial examples against the ResNet-18 model using the Logit, Logit-margin, and CFM methods. Then, we compute the cosine similarity (Sim.) between the latent features of the adversarial examples and the targeted class name. For reference, we also report the average targeted attack success rate (ASR) on the eight models. The results are depicted in table 1. We observe that as the similarity to the targeted class increases, the average attack success

Table 1: Evaluation on the cosine similarity (Sim) of the adversarial example features with its targeted attack class.

| Method | Logit | Logit-margin | CFM |
|--------|-------|--------------|-----|
| Sim.   | 22.7  | 22.8         | 24.2 |
| ASR.   | 23.1  | 23.8         | 38.6 |

Table 2: Transferable targeted attack success rate against various models. We respectively integrate the regularization term (C-) to three advanced targeted attacks methods, namely the Logit, Logit-margin, and CFM.

| Model | Method | RN-18 | RN-101 | RX-50 | DN-121 | ViT | PiT | Vis | Swin | Avg |
|-------|--------|-------|--------|-------|--------|-----|-----|-----|------|-----|
| RN-18 | Logit | **98.9** | 13.1 | 16.1 | 38.4 | **2.6** | 3.3 | 8.7 | 4.5 | 12.3 |
| | **C-Logit** | **98.9** | **13.7** | **19.1** | **41.9** | 2.3 | **3.6** | **9.3** | **5.8** | **13.7** |
| | Logit-margin | **100.0** | 15.8 | 19.5 | 42.7 | 2.5 | 3.6 | 9.3 | 5.1 | 14.1 |
| | **C-Logit-margin** | **100.0** | **16.8** | **20.4** | 41.5 | **5.5** | **6.6** | **13.4** | **8.1** | **16.0** |
| | CFM | 98.3 | 40.7 | 43.8 | 65.5 | 8.8 | 11.5 | 25.6 | 18.8 | 30.7 |
| | **C-CFM** | **98.4** | **42.1** | **47.1** | **70.1** | **12.1** | **14.8** | **30.8** | **22.2** | **34.2** |
| DN-121 | Logit | 19.7 | 12.4 | 17.0 | 98.4 | 1.5 | 3.0 | 6.6 | 2.7 | 8.9 |
| | **C-Logit** | **20.9** | **12.5** | **17.9** | 98.4 | **1.9** | **3.1** | **8.4** | **3.6** | **9.8** |
| | Logit-margin | **24.3** | 14.9 | 20.0 | 100.0 | 2.0 | 3.3 | 9.1 | 3.5 | 11.0 |
| | **C-Logit-margin** | 23.6 | **17.8** | **24.2** | 100.0 | **6.3** | **6.6** | **13.5** | **7.6** | **14.2** |
| | CFM | 78.7 | 64.0 | 70.0 | 98.0 | 21.4 | 28.2 | 49.7 | 34.5 | 49.5 |
| | **C-CFM** | **79.6** | **66.4** | **71.1** | 97.8 | **27.5** | **32.5** | **53.7** | **40.6** | **53.1** |
| ViT | Logit | 0.8 | 0.3 | 0.4 | 1.1 | 63.7 | 3.1 | 1.0 | 1.3 | 1.1 |
| | **C-Logit** | **1.0** | **0.8** | **0.8** | 1.1 | **64.6** | **5.3** | **2.1** | **1.9** | **1.9** |
| | Logit-margin | 0.5 | 0.8 | 0.5 | 1.1 | 75.2 | 4.2 | 1.2 | 1.7 | 1.4 |
| | **C-Logit-margin** | **1.5** | **1.2** | **2.1** | **2.2** | 60.5 | **6.3** | **3.8** | **3.8** | **3.0** |
| | CFM | 15.1 | 20.3 | 24.1 | 20.4 | 98.4 | 50.3 | 45.6 | 45.9 | 31.7 |
| | **C-CFM** | **20.5** | **29.5** | **33.2** | **32.2** | 97.8 | **60.1** | **52.9** | **51.0** | **40.0** |
| PiT | Logit | 0.0 | 0.4 | 0.0 | 0.8 | 0.1 | 85.8 | 1.3 | 1.1 | 0.5 |
| | **C-Logit** | **0.3** | **1.0** | **1.0** | **1.3** | **1.8** | **86.5** | **2.5** | **1.6** | **1.4** |
| | Logit-margin | 0.3 | 0.4 | **0.5** | 0.9 | 0.7 | 92.8 | 1.1 | 1.2 | 0.7 |
| | **C-Logit-margin** | **0.9** | **1.0** | **0.5** | **1.9** | **1.8** | 75.7 | **2.4** | **1.8** | **1.5** |
| | CFM | 5.3 | 9.1 | 12.3 | 9.1 | 15.2 | 98.6 | 29.6 | 27.1 | 15.3 |
| | **C-CFM** | **8.3** | **14.4** | **17.4** | **15.2** | **25.9** | **99.2** | **37.6** | **33.1** | **22.0** |

rates also improve. Notably, the targeted class information is modeled only by the target class name, without involving the visual modality. These results support our argument that: 1) *CLIP is a natural and effective indicator for evaluating black-box targeted adversarial transferability*; and 2) *the target class information can be modeled by different modalities in CLIP's latent space.*

## 3.3 Leveraging the CLIP to enhance the targeted transferability

We are motivated by the aforementioned findings to leverage CLIP as a helper to enhance targeted adversarial transferability. Recall that two factors contribute to the success of targeted attacks: amplifying the target class features and alleviating the original class features. Thus, we propose two terms to compute the distance of the current adversarial example to the target class and the original class based on CLIP.

Specifically, for the distance to the target class, we use the text embedding of "[Target class]" from CLIP to model the feature of the target class, then compute the cosine similarity between the adversarial example features and the text embedding as the distance. For the distance to the original class, we use the benign sample embedding from CLIP as the original class information and also use the cosine similarity to indicate the distance. The optimization can be formulated as follows:

$$\max \mathcal{L}_{reg} = \frac{E_{x^{adv}} \cdot E_{y^t}}{\|E_{x^{adv}}\|\|E_{y^t}\|} - \frac{E_{x^{adv}} \cdot E_x}{\|E_{x^{adv}}\|\|E_x\|}, \tag{2}$$

where $E_{x^{adv}}, E_{y^t}, E_x$ are the CLIP embeddings of the adversarial example, the target class name, and the benign sample, respectively.

**Results and insights**. We integrate the regularization term eq. (2) into Logit, Logit-margin, and CFM to form the C-Logit, C-Logit-margin, and C-CFM, respectively, and evaluate the targeted attack performance. The results are shown in table 2. For reference, we also report the average attack success rate (Avg) on the seven black-box models. There are three findings revealed by the results. First, the use of CLIP as guidance in targeted adversarial

Table 3: Transferable targeted attack success rate against various models. Four strategies are used in the mix-up operations of CFM attack, including the random (baseline), original, target, and combination.

| Model | Strategy | RN-18 | RN-101 | RX-50 | DN-121 | ViT | PiT | Vis | Swin | Avg |
|-------|----------|-------|--------|-------|--------|-----|-----|-----|------|-----|
| RN-18 | Random | 98.3 | 40.7 | 43.8 | 65.5 | 8.8 | 11.5 | 25.6 | 18.8 | 30.7 |
|       | Original | 98.1 | **41.5** | 42.8 | 65.4 | 9.1 | 12.1 | 27.9 | **19.0** | 31.1 |
|       | Target | **98.9** | 31.5 | 36.1 | 61.5 | 6.1 | 8.1 | 21.0 | 12.7 | 25.3 |
|       | Combination | 98.8 | 40.4 | **45.0** | **67.1** | 9.8 | 11.7 | 28.0 | 18.4 | **35.1** |
| DN-121 | Random | 78.7 | 64.0 | 70.0 | 98.0 | 21.4 | 28.2 | 49.7 | 34.5 | 49.5 |
|        | Original | 70.7 | 63.1 | 64.9 | **98.8** | **21.5** | 25.7 | 46.7 | 30.4 | 46.1 |
|        | Target | 63.3 | 44.6 | 51.0 | 98.0 | 8.7 | 12.9 | 28.7 | 16.0 | 32.2 |
|        | Combination | **72.3** | **65.9** | **66.5** | 98.6 | 19.7 | **26.9** | **47.3** | **32.5** | **47.3** |
| ViT | Random | 15.1 | 20.3 | 24.1 | 20.4 | 98.4 | 50.3 | 45.6 | 45.9 | 31.7 |
|     | Original | **7.8** | **16.8** | **18.8** | **13.6** | 98.0 | 49.7 | **36.1** | **31.8** | **24.9** |
|     | Target | 5.1 | 9.0 | 10.0 | 9.1 | 89.4 | 30.4 | 21.2 | 18.3 | 14.7 |
|     | Combination | 6.7 | 12.2 | 12.8 | 8.9 | 91.3 | 40.5 | 24.3 | 23.2 | 18.4 |
| PiT | Random | 5.3 | 9.1 | 12.3 | 9.1 | 15.2 | 98.6 | 29.6 | 27.1 | 15.4 |
|     | Original | **3.1** | **6.8** | **7.2** | **6.1** | **17.6** | 98.0 | **21.6** | **19.2** | **11.7** |
|     | Target | 0.9 | 1.5 | 1.0 | 1.1 | 2.3 | 88.8 | 4.6 | 4.2 | 2.2 |
|     | Combination | 1.5 | 4.5 | 4.5 | 3.6 | 9.1 | 92.5 | 10.3 | 11.8 | 6.5 |

attacks significantly boosts performance under the black-box setting, especially for CFM. It shows an improvement of up to 8.3% and 6.7% on average when using ViT and PiT as surrogate models. This phenomenon provides further evidence supporting the use of CLIP to model target class information. Second, the regions of interest used for classifying images into specific classes vary significantly among different models and architectures. We observe that the attack success rate on CNNs is very low when using a Transformer as the surrogate model, and vice versa. Additionally, the attack success rate under the white-box setting using the Transformer as the surrogate model doesn't reach 100% even with 300 iterations. While the untargeted attack success rate approaches 100% in recent studies, the research in targeted attack success rates remains heavily under-explored. Third, compared with the use of logit, the features are more helpful to craft targeted adversarial perturbation, where there is a clear performance gap between the CFM and Logit/Logit-margin.

## 3.4 Unraveling the Secrets Behind CFM's Success

From previous results, CFM stands out as the most effective targeted attack method, leveraging the features of benign samples to craft adversarial perturbations that approach the target class in the latent space. As discussed in the previous section, CFM employs a mixup of a limited set of images, neglecting specific information of the target class. Intuitively, in the context of transferable targeted attacks, one might expect that using more images belonging to the target class would yield better performance compared to using purely random images. But does this assumption hold true?

To answer this question, we set up four pool of images used for feature mix-up during the CFM attack, including 1) *Random*: we use random images for mix-up, which is the original implementation of CFM; 2) *Original*: we use the original image features for mix-up; 3) *Target*: we collect the images for each target class, and only the target class images are used for mixup during the attack; and 3) *Combination*: we randomly mix-up the target class image features with the original image features to achieve a good diversity of image features as well as involving more target features for better guidance.

**Results and insights**. We present the results in table 3. Contrary to intuition, using the target strategy does not improve targeted adversarial transferability; it even downgrades performance by an average of 13.2%. In comparison, the original strategy consistently achieves better performance, highlighting *the importance of fusing original benign sample features to boost performance rather than directly fusing target class features*. Additionally, we observe that the combination strategy, which mixes original image features with target class features, further improves targeted adversarial transferability when using CNNs as surrogate models. This indicates that *introducing target class information can enhance*

Table 4: Transferable targeted attack success rate against various models. We compare our proposed Fine-grained Feature Attack (FFA) with various advanced targeted attack methods, including SU, IDAA, AA, PoTrip, and CFM.

| Model | Method | RN-18 | RN-101 | RX-50 | DN-121 | ViT | PiT | Vis | Swin | Avg |
|---|---|---|---|---|---|---|---|---|---|---|
| RN-18 | SU | 99.5 | 7.0 | 8.0 | 21.7 | 0.2 | 0.4 | 2.7 | 1.2 | 17.6 |
| | IDAA | 87.5 | 3.3 | 3.7 | 12.7 | 0.1 | 0.6 | 1.9 | 1.3 | 13.9 |
| | AA | 4.8 | 0.7 | 0.7 | 0.8 | 0.3 | 0.1 | 0.1 | 0.2 | 1.0 |
| | PoTrip | 99.9 | 3.3 | 5.7 | 14.9 | 0.2 | 0.3 | 1.3 | 1.6 | 15.9 |
| | CFM | 98.3 | 40.7 | 43.8 | 65.5 | 8.8 | 11.5 | 25.6 | 18.8 | 30.7 |
| | **FFA** | **98.5** | **64.3** | **65.0** | **83.3** | **12.1** | **22.1** | **40.4** | **32.1** | **52.2** |
| DN-121 | SU | 16.3 | 9.8 | 12.1 | 99.3 | 0.4 | 0.5 | 4.3 | 1.6 | 18.0 |
| | IDAA | 15.5 | 6.8 | 9.5 | 90.2 | 0.4 | 1.9 | 3.5 | 2.9 | 16.3 |
| | AA | 0.6 | 0.2 | 0.1 | 78.1 | 0.0 | 0.0 | 0.0 | 0.0 | 9.9 |
| | PoTrip | 10.7 | 6.9 | 8.7 | 100.0 | 0.6 | 0.9 | 3.0 | 0.9 | 16.5 |
| | CFM | 78.7 | 64.0 | 70.0 | 98.0 | 21.4 | 28.2 | 49.7 | 34.5 | 49.5 |
| | **FFA** | **83.6** | **79.4** | **80.0** | **97.3** | **24.1** | **40.5** | **61.3** | **44.7** | **63.9** |
| ViT | SU | 0.7 | 0.9 | 0.7 | 0.8 | 39.9 | 3.4 | 2.5 | 2.1 | 6.4 |
| | IDAA | 2.6 | 2.6 | 4.0 | 3.8 | 35.4 | 8.6 | 5.8 | 6.2 | 8.6 |
| | AA | 0.0 | 0.1 | 0.0 | 0.2 | 29.7 | 0.0 | 0.0 | 0.0 | 3.8 |
| | PoTrip | 3.3 | 3.9 | 5.1 | 6.1 | 67.3 | 15.2 | 10.6 | 8.6 | 15.0 |
| | CFM | 15.1 | 20.3 | 24.1 | 20.4 | 98.4 | 50.3 | 45.6 | 45.9 | 31.7 |
| | **FFA** | **21.4** | **27.3** | **34.6** | **31.7** | **99.3** | **59.1** | **52.5** | **57.8** | **48.0** |
| PiT | SU | 0.5 | 0.7 | 0.5 | 0.5 | 0.5 | 76.4 | 1.7 | 1.4 | 10.3 |
| | IDAA | 1.7 | 2.3 | 3.4 | 2.6 | 2.0 | 48.3 | 7.1 | 7.3 | 9.3 |
| | AA | 0.3 | 0.1 | 0.0 | 0.3 | 0.0 | 10.7 | 0.0 | 0.0 | 1.4 |
| | PoTrip | 2.2 | 3.1 | 4.0 | 3.8 | 5.1 | 85.7 | 8.9 | 8.3 | 15.1 |
| | CFM | 5.3 | 9.1 | 12.3 | 9.1 | 15.2 | 98.6 | 29.6 | 27.1 | 15.3 |
| | **FFA** | **8.8** | **17.9** | **20.6** | **14.1** | **17.0** | **99.7** | **36.6** | **38.8** | **31.7** |

*targeted adversarial transferability without harming the original image features.* Among all experiments, the random strategy consistently achieves good performance across all surrogate models. This suggests that *feature diversity contributes the most to targeted adversarial transferability.*

## 3.5 Harnessing Fine-Grained Features for better performance

While previous findings have shown that "suitable guidance from the target class," *i.e.*, the "combination strategy," boosts targeted adversarial transferability when using CNNs as surrogate models, we also observe a significant performance drop when applying this insight to Transformer-based models. We attribute this to the differences in the working pipeline between CNNs and Transformers. While CNNs learn to detect objects from a global perspective, Transformers operate on the patch level. Compared to CNNs, Transformers can capture more fine-grained features, making the model more robust and mitigating the effectiveness of targeted adversarial perturbations crafted through global feature mix-up.

Specifically, rather than storing the features from the original images, we first partition them into multiple blocks and apply random input transformations to each block to further amplify the local features. Next, we

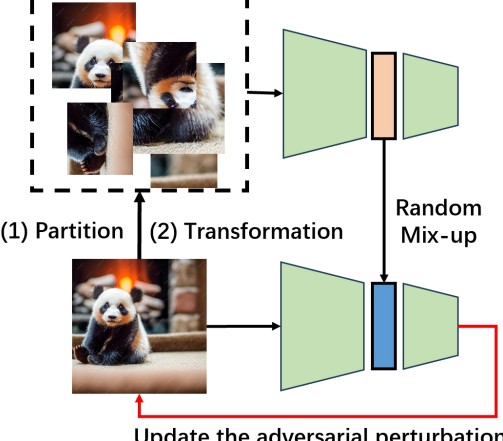

Figure 3: We leverage the fine-grained features to boost the targeted attack performance of feature-based attack, *i.e.*, the CFM.

forward these augmented images to the neural networks for feature storage. During attacks, the images are similarly enhanced by block transformations to highlight fine-grained features and are randomly mixed with the pre-stored fine-grained features. This design fully enhances feature diversity and introduces more competition among leveragable features, thereby boosting targeted adversarial transferability.

**Results**. We present the results in table 4. It can be shown that the FFA consistently achieves the state-of-the-art performance against different models, with a clear gap of $xx.xx\%$. It sufficiently supports our argument that *paying more attention on fine-grained features could boost the targeted adversarial transferability*. It should be also noted that, though proposed method is effective on attacking CNNs (some of results even achieve nearly 90% targeted attack success rate), there remains a room for improving the performance on Transformers.

## 4 CONCLUSION

In this work, by studying three advanced targeted adversarial attack methods, we derive the general insight that modeling the target class information suitably can significantly boost targeted adversarial transferability. We empirically find that CLIP serves as an excellent indicator for modeling target class information, enhancing attack performance. We also delve into feature-based attacks to uncover underlying principles in deisgning an efficient targeted attack, including the careful design of mix-up strategies and the importance of feature diversity. Furthermore, we propose leveraging fine-grained features to improve targeted adversarial transferability. Extensive experiments on the ImageNet-1K dataset, along with various defense models and commercial APIs, robustly demonstrate the effectiveness of our proposed method.

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
