# OpenReview forum: "Towards good practice in boosting the targeted adversarial attack"
_ICLR.cc/2025/Conference — Submitted to ICLR 2025_

### Official Review · Reviewer_QCGN · 2024-10-28

**Soundness:** 3
**Presentation:** 1
**Contribution:** 2
**Rating:** 5
**Confidence:** 3

**Summary:**

The paper proposes a series of enhancements to targeted adversarial attack methods, focusing on boosting the transferability of attacks in a black-box setting.  The paper identify that CLIP is a good indicator to evaluate a good transferable targeted attacks and propose a regularization term to boost the existing transferable targeted attacks.

**Strengths:**

1. Effective Enhancement Methods: The proposed method regularization term is simple and effective.

2. Comprehensive Experiments: Extensive experimentation on a wide range of models, including CNNs and Transformers.

**Weaknesses:**

1. Poor presentation: The presentation of this paper is poor and make it hard to read. In particular, methods, experimental settings, and experimental results are all mixed in one section, which makes reading extremely difficult. At the same time, the method part lacks formal definitions.(i.e in sec 3.5 The text does not indicate the complete definition of FFA but it appear in the table 4 and sec 3.5 results)

2. The description of Methodology is not detailed: it lacks the necessary formulaic description for method in 3.4 and 3.5.  Some parts of the methodology, particularly the regularization term formulation and the fine-grained feature enhancement, could benefit from clearer explanations and theoretical justifications

3. Lack of theoretical analysis: the paper does not provide a formal theoretical foundation analysis on proposed methods.For example, the reasons for the effectiveness of the fine-grained feature partitioning or the specific formulation of the regularization term are not well-justified theoretically.

4. Lack of efficiency comparsion:  I'm also concerned about the efficiency of the adversial method.  Since the author has introduced CLIP will bring extra overhead. The author should add efficiency analysis to the proposed methods to enhance soundness.

5. Lack of necessary ablation study: The ablation experiment is needed to further present the robustness of proposed methods. For example, How sensitive is the performance to different hyperparameter choices (e.g., regularization strength, feature partitioning granularity?

6. (minor) typos appear in the paper: in line 176 "white-". In line 428, with a clear gap of"xx.xx%".

**Questions:**

1. Previous work such as CFM were all conducted on the ImageNet-Compatible dataset. However, the author follow this standard setting, please clarify the reasons.

2. Adversarial training is a common defense. How does the regularization term affect attacks on models trained with strong adversarial training techniques?

Other questions see weakness.

---

### Official Review · Reviewer_SXQ5 · 2024-11-03

**Soundness:** 2
**Presentation:** 1
**Contribution:** 1
**Rating:** 3
**Confidence:** 4

**Summary:**

The paper explores improving targeted adversarial attacks in black-box settings by leveraging the CLIP model to enhance transferability. By using CLIP’s joint vision-language representations, the authors introduce a method to align target class information between surrogate and victim models. Their approach includes a new metric based on CLIP for evaluating attack effectiveness and a regularization term that boosts targeted attack success by focusing on fine-grained features. Experiments on ImageNet-1K demonstrate that their method significantly improves attack success rates, especially on CNN-based models, highlighting CLIP's potential to enhance black-box targeted adversarial attacks.

**Strengths:**

- The paper introduces a method that leverages CLIP to improve the success rate of targeted adversarial attacks in black-box settings, addressing a challenging aspect of adversarial machine learning. The experimental results shows the effectiveness of the proposed method.

**Weaknesses:**

- The presentation falls short of ICLR standards, with some aspects lacking mathematical rigor. For instance, in Table 1, cosine similarity is shown on a range from (-100, 100), whereas it is conventionally bounded between (-1, 1), which could lead to confusion or misinterpretation. Additionally, in Equation (2), the "max" operation is used on the left side without indicating the variable being optimized, creating ambiguity and making the equation appear incomplete or improperly formulated.

- The paper reclaims several already established conclusions. For example, the finding in Section 3.2 that targeting an image to a specific class in one model increases the probability of it being classified to that target class in another model (in this case, CLIP) is a well-known property of adversarial attacks, known as transferability. This section devotes significant content to extending an established concept to CLIP, which seems unnecessary and does not add meaningful value to the study.

- The proposed method is relatively straightforward, as it primarily adds CLIP as an additional regularization objective to conventional black-box attacks. This essentially equates to introducing a stronger surrogate model to enhance the attack. The reliance on CLIP for embedding alignment, without leveraging its multimodal capabilities, limits the novelty of the approach. This raises questions about whether a similar performance boost could be achieved by using other pretrained classification models, making the choice of CLIP less compelling.

**Questions:**

Does the author try to use another pretrained classifier to replace CLIP as a surrogate model for regularization?

**Details Of Ethics Concerns:**

The paper proposes an adversarial attack methods that may mislead machine learning system.

---

### Official Review · Reviewer_V3nX · 2024-11-03

**Soundness:** 2
**Presentation:** 1
**Contribution:** 2
**Rating:** 3
**Confidence:** 4

**Summary:**

This paper presents a novel transferable targeted adversarial attack for image classification models.

The authors introduce a CLIP-based metric, asserting that it effectively indicates the transferability of targeted attacks. Building on this, they propose a CLIP-based regularization approach to amplify features of the target class while diminishing those of the original class, leveraging CLIP’s capacity as a large pre-trained model to capture class features.

The paper further demonstrates that a random patch strategy can enhance the existing attack method, CFM.

Overall, the proposed method, FFA, outperforms CFM [CVPR 2023] in targeted attack transferability.

**Strengths:**

- The overall boost in targeted attack transferability of the proposed FFA is large, as in Table 4.
- The proposed CLIP regularization is simple and practically easy to implement.

**Weaknesses:**

- [W1] The presentation is not good
    - [W1.1] The authors use the wrong font for ICLR.
    - [W1.2] The CLIP variant that was used is not described. This is important because using CLIP based on ViT or ResNet makes a large difference in interpreting the results. Also, the model size can matter.
    - [W1.3] The citations for the existing methods used in Table 4 are unclear. The abbreviations should be referred to in the main text.
- [W2] The effectiveness of the proposed CLIP-based metric to predict the effectiveness of transferable adversarial attacks has only been weakly supported using three methods, as shown in Table 1. Why not calculate the metric on other adversarial attack algorithms in Table 4?
- [W3] The authors’ claim that “the key to successful transferable targeted attacks” is to “amplifying the target class features and alleviating the original class features” has not been verified well.
    - The authors should provide an ablation study of the regularization term (Eq.2), using only the first, second, and both terms.
- [W4] Adding CLIP regularization is nearly equivalent to using an additional surrogate model. Therefore, I do not think Table 2 is a fair comparison between methods (i.e., “CFM” uses only one surrogate model, but “C-CFM” uses two surrogate models).
    - I expect the authors to compare, for example, “CFM with using an ensemble of DN-121 and ViT” versus “C-CFM using DN-121 and a CLIP-based regularization” so that the number of models used for adversarial attacks are fixed to two.
- [W5] Sec.3.4 and 3.5 is not well organized.
    - [W5.1] How are these sections connected to previous sections of using CLIP? I did not understand the connection between them.
    - [W5.2]  Table 3 shows inconsistent results between CNNs and ViTs. The claim that “introducing target class information can enhance targeted adversarial transferability without harming the original image features” seems not verified well.
    - [W5.3]  The algorithm is not described well. What is the exact type of transformation and the number of transformations? What is a random mixup? Is CLIP regularization used here?

**Questions:**

Please see the weakness.

---

### Official Review · Reviewer_RwEX · 2024-11-04

**Soundness:** 2
**Presentation:** 3
**Contribution:** 1
**Rating:** 3
**Confidence:** 4

**Summary:**

Constructing transferrable targeted adversarial attacks in black-box setup is a challenging task, as the target model could have a different classification boundary from the surrogate model used to generate the attack. This paper aims to craft attacks that are more likely to transfer. Based on the observation that foundation models like CLIP are trained on a large-scale dataset, thus are more likely to capture better features, the authors proposed two new regulation terms to guide the generating adversarial attacks: one term measures the cosine similarity between the visual encoding of the original clean sample and the current input; the other term measures the cosine similarity between the visual encoding of the current input and the text encoding of the target class. The objective is to maximize the first distance while minimizing the second distance. Evaluation shows these two terms can boost the transferrability of black-box attacks for three different attack methods.

**Strengths:**

+ The proposed method indeed showed improved performance for three attacking methods

**Weaknesses:**

As mentioned in the related work section:

> Springer et al. (2021) find that a surrogate model that is more robust to adversarial perturbation can be leveraged to craft adversarial examples with highly targeted adversarial transferability.

As a foundation model trained on large-scale dataset, CLIP is likely to be more robust against adversarial perturbations (e.g., perturbations generated on ResNet is less likely to transfer to CLIP). Based on this observation/theory, adversarial examples crafted based on CLIP is more likely to transfer. This may explain the success in the experiments presented in the work. So, a basic question is, if we directly use CLIP as the surrogate model to generate attacks (e.g., based on the two terms proposed in the paper), how well would the generated attacks transfer? If they transfer well, what's the contribution of this work?

**Questions:**

Please including a new baseline that uses CLIP as the surrogate model and the two proposed terms as objectives to generate adversarial examples.

**Details Of Ethics Concerns:**

The paper presents a better attacking method to fool DNN-based image classifiers.

---

### Meta-Review · Area_Chair_gGRZ · 2024-12-21

**Metareview:**

3x reject, 1x borderline reject. This paper introduces a targeted adversarial attack method leveraging CLIP-based metrics and regularization to improve black-box transferability. The reviewers agree on the (1) relative simplicity of CLIP-based augmentation, (2) observed boost in success rates against CNN-based victim models, and (3) comprehensive experiments across multiple surrogates. However, they note (1) poor organization and missing methodological details, (2) unclear novelty beyond using CLIP as a second surrogate, and (3) insufficient ablations, theoretical analysis, and efficiency comparisons. No follow-up responses were provided, and these concerns remain unaddressed, so the AC leans to not accept this submission.

**Additional Comments On Reviewer Discussion:**

N/A

---

### Decision · Program_Chairs · 2025-01-22

Reject